# Enabling stable preservation of ML algorithms in high-energy physics with petrifyML

*A. Buckley[1], L. Corpe[2], M. Habedank[1], T. Procter[3]*

[1] *School of Physics & Astronomy, University of Glasgow, G12 8QQ, Glasgow, UK*
[2] *Université Clermont Auvergne, Laboratoire de Physique de Clermont Auvergne, CNRS/IN2P3, Clermont-Ferrand, France*
[3] *Jagiellonian University, Łojasiewicza 11, 30-348 Kraków; Poland.*

**Abstract**

Machine learning (ML) in high-energy physics (HEP) has moved in the LHC era from an internal detail of experiment software, to an unavoidable public component of many physics data-analyses. Scientific reproducibility thus requires that it be possible to accurately and stably preserve the behaviours of these, sometimes very complex algorithms. We present and document the petrifyML package, which provides missing mechanisms to convert configurations from commonly used HEP ML tools to either the industry-standard ONNX format or to native PYTHON or C++ code, enabling future re-use and re-interpretation of many ML-based experimental studies.

## 1 Introduction

Despite being early adopters of machine learning (ML) since the 1980s [1, 2], high-energy physics (HEP) experiments until recently typically limited its use to "internal" mechanisms such as physics-object calibration and reconstruction rather than directly publishing ML outputs as scientific results. The classic example of this is the flavour "tagging" of hadronic jets [3, 4], which has long been ML-based, and more recently other complex reconstructions such as electron/photon or tau/hadronic jet disambiguation where ML methods can improve performance over closed-form calibrations. In this mode, ML effects could largely be summarised as simple tabulations of object reconstruction efficiencies or resolutions, so re-interpretation of HEP experimental publications did not require re-running the original trained ML algorithm.

The Run 2 phase of the Large Hadron Collider has changed this picture considerably, coinciding as it has with the rapid rise of both well-aligned computing architectures and new, highly effective ML architectures. These effects have been transformational in HEP: not only have modern architectures and computing power produced step-changes in the performance of tagging and other algorithms, but they have also underpinned the development of data-analysis techniques which provide non-parametrically optimal sensitivity to proposed models of new physics. While excellent news for the ability of experiments to study given (and typically rather simplified) models, this foregrounding of ML methods as an integral part of the physics study itself – not replaceable with an efficiency tabulation – introduces technical challenges for scientific reproducibility, particularly in the long term, which have only recently begun to receive significant attention [5].

These problems centre on toolkit dependencies and format stability. Current ML algorithms are typically trained, and often deployed, from PYTHON-based tools, but these are then not easily usable from other computing languages. Even for those who wish to re-use from PYTHON, the variety of toolkits is large and a particular preserved ML algorithm may behave differently even in different versions of its toolkit. This, and the desirability of transfer between different training and runtime environments, motivated the development and evolution of the ONNX format [6]. However, several ML frameworks – in particular tools developed specifically within HEP, such as TMVA [7] and `lwtnn` [8] – do not natively support conversion to ONNX. Concerns remain about the long-term reproducibility of ONNX behaviours as their execution environments evolve, without cumbersome and slow approaches such as wholly containerising each trained ML's runtime environment: this motivates also, in some cases, preserving algorithms as dependency-free native code.

The petrifyML tool described in this release note was created to fill gaps in this chain of algorithm preservation. A PyPI-installable PYTHON package and set of command-line tools, it provides converter routines to ONNX for neural nets (NNs) created with the `lwtnn` and TMVA HEP frameworks, and to ONNX and native C++ and PYTHON for boosted decision trees (BDTs) created with TMVA and `scikit-learn`.

## 2  Installation and dependencies

petrifyML is fully written in PYTHON and strives to depend only on a minimal suite of well-maintained, standard, and lightweight packages. Naturally, these encompass ONNX and OnnxRuntime [9], with `numpy` [10] and `pandas` [11] used for utilities. TensorFlow [12], `tf_keras` [13], and `tf2onnx` are used for the conversion of TMVA multilayer perceptrons via Keras. `uproot` [14] is used for converting `MVAUtils` BDTs, and the heavyweight ROOT framework is only required for conversions from TMVA BDTs to plain-text C++ or PYTHON. Lastly, `pytest` [15], `pytest-cov`, and `pyyaml` are employed for the automated testing of the conversion routines.

All scripts optionally output an example validation script or similar. Unavoidably, these validation codes have a larger set of dependencies. For example, converting an `MVAUtils` ROOT file to ONNX requires neither ROOT nor OnnxRuntime, but the validation macros require both ROOT (with an `MVAUtils` install) and OnnxRuntime.

The petrifyML package (and its dependencies) can be conveniently installed using `pip`:

```
$ pip install petrifyml[part]
```

Specifying a `[part]` allows to have a lightweight installation with only the required packages for a single operation mode of petrifyML: "sklbdt", "tmvabdt", "lwtnn", "tmvamlp", "mvautils", or "dev" for all parts and the test suite.

## 3  Converting BDTs to plain code C++ and Python

Conversion to native C++ and PYTHON code avoids any sort of dependency, guaranteeing verbatim performance in perpetuity for smaller BDTs. This is particularly advantageous as the industry focus driving the development of the ONNX format is on short-timescale shifts between platforms rather than long-term preservation. This means that the long-term reproducibility of ONNX behaviours as their execution environments evolve need to be monitored. Conversely, native, human-readable C++ and PYTHON code grow quickly to a considerable size, limiting this conversion to small and medium-sized BDTs for practical use.

### 3.1  petrify-sklbdt-to-cpp

*Install with* `pip install petrifyml[sklbdt]`

#### 3.1.1  Background

The `scikit-learn` [16] PYTHON package provides a variety of models for decision trees and forests. The script described in this section is designed for the `GradientBoostingClassifier`, though should be readily extensible to other formats if the need arises. `scikit-learn` classifiers are normally saved in PYTHON's pickle format. Given that this format can prove highly version dependent and unstable over even short periods of time, saving a pickle file (absent an entire container with the right dependencies installed) cannot be considered model preservation. Therefore, there is a clear need for a reliable, long-term preservation option for these BDTs.

#### 3.1.2  Running

The script accepts trees in either `.pkl`[1] or `.job` formats. For a classifier stored in a `.pkl` file, a simple use case would be

```
$ petrify-sklbdt-to-cpp mytree.pkl -n my-cpp-file
```

---

[1] Note that PYTHON's pickle format is generally not backward compatible; ensure to save using a pickle protocol version supported by the PYTHON version used to run petrifyML.

Note that the `.cc` suffix is appended automatically to the `-n` argument. The script also supports input in `.job` format.

Further information on all optional arguments can be accessed via the `-h` option; most are self-explanatory, but some deserve some additional notes:

- `--write-validation` (default: `False`): Appends a `main` function to the output `.cc` file that performs test calls of the BDT function on randomly generated input, and prints compilation instructions, in order to allow easier validation.

- `--run-validation` (default: `False`): As above, but also compiles the new `.cc` file, runs it, and compares output to the original `scikit-learn` version. Assumes that the `g++` compiler is available. Optionally, if a list of numbers is provided as the argument, these are used as the input to the BDT instead of randomly generated inputs (in this case it is the user's responsibility to ensure they provide the correct number of inputs).

## 3.2   petrify-tmvabdt-to-cpp and petrify-tmvabdt-to-py

*Install with* `pip install petrifyml[tmvabdt]`

### 3.2.1   Background

The Toolkit for Multivariate Data Analysis (TMVA) is the ROOT library that provides interfaces and implementations for ML models. This includes a variety of BDTs (including multi-class classifiers), which can be stored as either XML or ROOT files. These formats are unfortunately tied to the heavyweight ROOT package, and therefore there is a motivation to be able to provide a lightweight alternative for simple trees. This comes in the form of converters to plain-text C++ or PYTHON.

### 3.2.2   Running

For a classifier stored in an XML file (the converter does not take ROOT files), a simple use case would be

```
$ petrify-tmvabdt-to-cpp mytree.xml -n my-cpp-name
```

or

```
$ petrify-tmvabdt-to-py mytree.xml -n my-py-name
```

Note that the appropriate `.cc` or `.py` suffix is appended automatically to the `-n` argument.

Further information on all optional arguments can be accessed via the `-h` option; most are self-explanatory, but some deserve some additional notes:

- `--write-validation` (default: `False`): Appends a `main` function to the output `.cc` or `.py` file that performs test calls of the BDT function on randomly generated input. Prints compilation instructions in the C++ case, in order to allow easier validation.

- `--run-validation` (default: `False`): As above, but also runs the new `main` function and compares output to the original TMVA version. In the C++ case, this option assumes that the `g++` compiler is available. Optionally, if a list of numbers is provided as the argument, these are used as the input to the BDT instead of randomly generated inputs (in this case it is the user's responsibility to ensure they provide the correct number of inputs).

- `--nClasses`, `--nc` (default: -1): See Section 3.2.3.

### 3.2.3   Multi-class

BDTs are not necessarily limited to binary classification problems, and TMVA supports multi-class BDTs. In order to petrify a multi-class classifier, the user must supply the (correct) number of output classes via the `--nClasses` argument. Supplying an incorrect number (or failing to provide one, while feeding in a multi-class XML file) will lead to undefined behaviour. It is likely that a C++ or PYTHON function will still be written "successfully": however if the number of classes provided is an under-estimate this function will likely crash; and if it is an over-estimate then the returned result will not be correct.

# 4   Converting BDTs to ONNX

ONNX files are increasingly the universal standard for ML preservation, come with the blessing of the Les Houches guidelines on reinterpretable ML [5], and are already supported by most major reinterpretation frameworks. Though in principle multiple programs can be used to carry out inference using the information in an ONNX file, in practice the most commonly used (including in the RIVET, COLLIDERBIT and CheckMATE ONNX interfaces) is OnnxRuntime [9].

## 4.1   petrify-mvautilsxgboost-to-onnx and petrify-mvautilslgbm-to-onnx

*Install with* `pip install petrifyml[mvautils]`

### 4.1.1   Background

`MVAUtils` [17] is a package developed by the ATLAS collaboration for the preservation and portability of BDTs. Notably, `MVAUtils` does not train BDTs itself: rather it takes BDTs that have been trained and stored in custom formats by diverse tools (`xgboost` [18], `lgbm` [19] and TMVA), and converts them into a ROOT-based [20] format (which does differ based on the original training environment) which is typically significantly more compressed.

The objectives of the `MVAUtils` package share some overlap with those of this package, and readers may ask why, if the work has already been carried out to support conversion to a format that can be read by one universal package, is another project needed? Indeed, the CheckMATE2 [21] reinterpretation tool already uses parts of the `MVAUtils` code (released as part of the Athena [17] framework) to run BDTs published by ATLAS.

However, the dependence on ROOT is an insurmountable issue for many other tools, such as RIVET, and GAMBIT for full production level scans. ROOT significantly complicates compilation, can cause issues with multithreading and may be impractical for users to run locally on their own machines. Therefore, the separate convertor presented in this release note fills an uncovered hole.

As BDT forests get larger, writing them in plain-text code files as in Section 3 may become impractical. For example, the `MVAUtils` forests used by ATLAS in the higgsino search in References [22, 23] include 125,000 trees, each with 120-130 nodes. At a conservative estimate, this would require a C++ file over 30 million lines long, which is impractically large for a nominally "lightweight" format.

The alternative, implemented here, is to convert the trees from various proprietary or "public-yet-impractically-difficult-to-reuse" (sadly common in HEP) formats into ONNX files.

At the moment, petrifyML only provides converters for `MVAUtils` files created from `xgboost` and `lgbm` forests. The main reason for this is need: ATLAS has not (yet) published any `MVAUtils` ROOT files that originated in TMVA format. The converter uses the `pip`-installable `uproot` package to parse the input files (a ROOT installation is *not* required to run this converter[2]), and directly constructs the BDT using the ONNX TreeEnsemble operator without intermediate steps.

If the original `lgbm` or `xgboost` trees are available, then it is of course preferable to convert to ONNX using the inbuilt methods of those packages: we do not intend this package to become part of a baroque chain of conversions. However, it is regrettably common for the original files to be either not publicly released or indeed be completely lost.

### 4.1.2   Running

A simple use-case for both input formats is shown below:

```
$ petrify-mvautilslgbm-to-onnx myfile.root -n my-onnx-file

$ petrify-mvautilsxgboost-to-onnx myfile.root -n my-onnx-file
```

Note that the `.onnx` suffix is automatically appended to the `-n` argument.

Further information on all optional arguments can be accessed via the `-h` option; most are self-explanatory, but some deserve some additional notes:

- `--nf`, `--nfeatures` (default: `None`): The number of input features for the forest. If it is not provided, the script will try and guess based on the first few trees: however the nature of the `MVAUtils` format means that this guess can be wrong, particularly if there is a feature that is not

---

[2] Although if users wish to rerun `MVAUtils` for comparative validation, a full installation of ROOT is unavoidable.

used in the first few trees of the forest[3]. If the guess is wrong, then this will not be apparent until inference fails. SimpleAnalysis [26] or similar pseudocode may be the best source of this information.

- `--classifier` (default: `False`): When BDTs are used as classifiers, it is customary to apply a sigmoid function to convert the BDT-score into a (pseudo-)probability. In the `MVAUtils` format, this information is not included in the ROOT file, but rather is dependent on whether the BDT is called with `GetClassification` or `GetResponse`. While the user is free to replicate this themselves, if the network is only ever to be used as a classifier, it is simpler if this function is handled within OnnxRuntime. Therefore, running with `--classifier` applies a final sigmoid function within the ONNX description of the model.

- `--write-validation` (default: `False`): See Section 4.1.3.

- `--opset` (default: `None`) and `--ir-version` (default: `None`): See section 4.1.4.

### 4.1.3 Validation

The `--write-validation` option will, after the successful conversion of the network, write a directory containing four files:

- `testonnx.py`: A python script that runs the ONNX implementation of the BDT on a set of inputs, with the result printed to screen and saved to `testonnx.csv`. It is the user's responsibility to make sure the OnnxRuntime PYTHON module (not needed for the conversion itself) is available.

- `testmvautils.cxx`: A ROOT macro that runs the original `MVAUtils` implementation of the network, on the same set of inputs as `testonnx.py`, also printing the result to screen and also saving it to `testmvautils.csv`. Access to `MVAUtils` is the user's responsibility (e.g. via lxplus or an Athena docker image).

- `validator.py`: A minimal PYTHON script that checks that the two `.csv` files agree.

- `readme.md`: Brief documentation reminding users how to use the scripts in the validation directory.

As with the `lwtnn` validation described in Section 5.2.3, this is not a comprehensive check, but will likely catch most issues which are not pathological; and can serve as a starting point for further validation. For BDTs in particular, it may be more useful to replace the randomly generated inputs in `testonnx.py` and `testmvautils.cxx` with numbers at a more physically relevant scale.

### 4.1.4 Version compatability options

The petrifyML implementation of boosted decision trees uses the `TreeEnsemble` operator from the `ai.onnx.ml` domain. This operator was only introduced in version 1.16 of ONNX (March 2024), and supported in OnnxRuntime from version 1.18.0 (May 2024). Therefore, for inference to work, it is important to be using a (relatively) up-to-date version of OnnxRuntime. While this is unlikely to be problematic in industry, the adoption of up-to-date package versions has not always been straightforward in HEP.

As these features are still somewhat fresh in ONNX and OnnxRuntime, forests produced using cutting edge ONNX may not always be interpretable by even only slightly older (but possibly still ≥ 1.18.0) versions of OnnxRuntime. For example, running petrifyML with its default settings using ONNX 1.18 produces ONNX files that cannot be loaded by OnnxRuntime 1.22.0[4], due to mismatches in the default ONNX `opset` and intermediate-representation versions.

Therefore, we allow users to specify both the opset and intermediate-representation version via the `--opset` and `--ir-version` options[5].

There are no backward compatibility issues, so trees produced with older ONNX `opset`s should remain usable in all future releases of OnnxRuntime. Not specifying either variable (or explicitly setting to `None`) defaults to the most recent versions supported by the installation of ONNX that petrifyML uses.

---

[3] Indeed, for one of the forests released alongside References [24, 25], the final input feature was *never* used and hence totally invisible in the `MVAUtils` format.

[4] Notably, for a period of time these were the latest releases of either package.

[5] If unsure which versions are compatible with your installation, `opset` 22 and `ir-version` 10 should be a reliable base.

# 5 Converting Neural Nets to ONNX

All the same arguments that apply to preserving BDTs in ONNX apply to preserving neural nets in ONNX. If anything, ONNX is an even stronger industry standard with respect to neural nets.

## 5.1 petrify-tmvamlp-to-onnx

*Install with* `pip install petrifyml[tmvamlp]`

### 5.1.1 Background

TMVA, ROOT's library for interfaces to and implementations for ML, in addition to the aforementioned BDTs provides the option to train multilayer perceptrons (MLPs). Their weights are stored in ROOT or XML files but are unfortunately tied to the heavyweight ROOT package. With the System for Optimized Fast Inference code Emit (SOFIE), TMVA was extended to allow reading ONNX files produced in other ML libraries. Notably, SOFIE does *not* allow TMVA to *write* ONNX files such that they could be used in other ML libraries.

This shortcoming is bridged by petrifyML. Technically, petrifyML reads the XML file, creates an MLP in TensorFlow [12] using Keras [13] with the same architecture and node weights, and exports it to ONNX files using `tf2onnx` [27].

### 5.1.2 Running

A simple use-case is:

```
$ petrify-tmvamlp-to-onnx my-tmvamlp-file.xml -n my-onnx-file
```

Further information on all optional arguments can be accessed via the `-h` option; most are self-explanatory, though some deserve additional notes:

- `--run-validation` (default: `False`). Validates that using the original XML file with TMVA, the generated Keras model, and the generated ONNX file with the OnnxRuntime yield the same inference results on test data. The test data can be given with `--test-data`. If not given, it is random-uniformly generated within the parameter ranges allowed for each input variable.

- `--validation-only` (default: `False`). Only runs the validation, i.e. does not generate a Keras model or output an ONNX file. Allows to run the validation quickly and repeatedly.

## 5.2 petrify-lightweightnn-to-onnx

*Install with* `pip install petrifyml[lwtnn]`

### 5.2.1 Background

`lwtnn` [8] is a package designed for fast and efficient NN inference in C++ environments with minimal dependencies, even when the original network was trained in an entirely PYTHONic environment, for example using Keras [13]. It was originally designed for use in the ATLAS trigger, but particularly during LHC Run 2 and before the usage of ONNX became more widespread, it became a common way for ATLAS analyses and tagging groups to run neural net inference inside C++ analysis frameworks.

In many ways, this is a very good format for reinterpretation: it has minimal dependencies, uses human-readable files and is easy to link from other C++ projects. Indeed, RIVET has had a header-only interface to `lwtnn` since version 3.1.7, which allows individual analysis plugins to be compiled against `lwtnn` as required. Nevertheless as `lwtnn` usage wanes even within ATLAS – due to a combination of the emergence of ONNX and `lwtnn`'s lack of support for cutting-edge architectures – it will be hard for all reinterpretation tools to justify linking against an aging package that may only be required for one or two analyses: thus motivating a method of converting to ONNX. This will also allow models previously only published in `lwtnn` form to be used in PYTHON for the first time.

`lwtnn` stores neural net weights in a `json` format. The exact `json` schema depends on the network architecture: `lightweightneuralnetwork` for simpler linear models; and `lightweightgraph`, for more complex models. At the moment, only conversion of the former is supported in petrifyML. Similarly, as summarised in Table 1, only a subset of the layer-types and activation functions that

`lightweightneuralnetwork` supports are currently supported for conversion to ONNX. Nevertheless, the present coverage is sufficient to cover all `lwtnn json` files so far publicly released by ATLAS[6]. Hopefully, coverage will be extended with time, especially if the need emerges due to ATLAS publishing more `lwtnn` files. The petrifyML code is also sufficiently modular though that a moderately technically experienced user should be able to add their own layers and activation functions if required[7].

| Supported `lwtnn` layer types | Supported `lwtnn` activation functions |
|:---:|:---:|
| Dense | Relu |
| Normalization | Sigmoid |
| | Softmax |
| | Elu |
| | Tanh |

Tab. 1: `lwtnn` layer types and activation functions currently supported when converting `LightweightNeuralNetwork` to ONNX.

### 5.2.2 Running

A sufficient example for most use-cases is:

```
$ petrify-lightweightnn-to-onnx my-lwtnn-file.json -n my-onnx-file
```

Further information on all optional arguments can be accessed via the `-h` option; most are self-explanatory, but some deserve some additional notes:

- `--float-type` (options: `float, float16, double`, default: `float`). This changes the data-type that the ONNX file expects as input and provides as output. This is particularly relevant to `lwtnn` conversion as `lwtnn` uses `double`'s internally, so if you wish to recreate the original `lwtnn` score close to or beyond single floating-point precision, this may be an option you wish to specify. However, note that 32-bit `float`s are such a common default for ONNX files that some interfaces to OnnxRuntime in HEP codes may have `float` hard-coded.

- `--opset` (default: `None`) and `--ir-version` (default: `None`): As described for `MVAUtils` in section 4.1.4, although given the maturity of the ONNX operators used for simple DNNs, using very recent `opset`s and intermediate-representation versions is less likely to be important in this case.

- `--write-validation` (default: `False`): See Section 5.2.3.

### 5.2.3 Validation

The `--write-validation` option will write a C++ file that will carry out inference using both the original version of the `lwtnn` network and the new ONNX version, comparing the result. As it is only evaluated on one set of input features, it is not a comprehensive validation, but it is a useful first check, and will provide much of the tiresome boiler-plate for the conscientious but potentially time-strapped user who wishes to write a more comprehensive validation script. It is the responsibility of the user to make sure both `lwtnn` and OnnxRuntime (neither of which are requirements to run the converter) are correctly installed and linked. The input features are randomly generated, using the normalisation information contained within the `lwtnn json` file to produce inputs at a "physically reasonable" scale.

## 6  Summary

We have introduced the petrifyML package for converting configurations from commonly used HEP ML tools to either native PYTHON or C++ code, or the industry-standard ONNX format. The package has few and lightweight dependencies. In combination with the easy installation via `pip` and straightforward command-line interfaces, this makes it well suited to simplify the long-term preservation of HEP ML models.

---

[6] At the time of writing, this comes to a grand total of one neural network [28] from an ATLAS search for vector-like quarks [29].

[7] Merge requests will always be welcome!

At the time of writing, the package can convert all non-standard model formats published by the AT-LAS collaboration without native ONNX support. Future developments might extend the functionality to further `MVAUtils` and `lwtnn` models, if the need arises.

## 7    Acknowledgements

AB, MH and TP acknowledge funding from OpenMAPP project, via UKRI/EPSRC EP/Y036360/1 and National Science Centre, Poland under CHIST-ERA programme (NCN 2022/04/Y/ST2/00186).

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
