# Peer review of "Enabling stable preservation of ML algorithms in high-energy physics with petrifyML"

_SciPost Physics Codebases_

## Round 1 · Referee Report · Louis Moureaux (Referee 1) · 2025-9-29

Strengths
1- The presented software fills a gap preventing the reinterpretation of results obtained with machine learning models, when the models were developed using certain libraries widespread in high-energy physics. 2- An approach with minimal dependencies is proposed, enabling the low-maintenance long-term preservation of models and ensuring equal availability across operating systems and CPU architectures. 3- The documentation is clear and aims at comprehensiveness. 4- The code base is clear and good practices are followed during development.
Weaknesses
1- No benchmarking test is provided. 2- No example application is presented. 3- The overlap with other tools has been overlooked, especially emlearn. 4- The interface of the emitted code is not documented. 5- Legal questions about the emitted code should be clarified.
Report
The submission documents PetrifyML, a conversion tool for the long-term preservation of machine learning models used in high-energy physics. PetrifyML can emit source code or (depending on the input format) ONNX files for boosted decision trees originally implemented in various generic and domain-specific frameworks. It also supports simple neural networks. For most of the supported use cases, PetrifyML is the only available solution to convert the models to a stable format usable for reinterpretation.
The software's unique feature set makes an excellent case for publication in SciPost Physics Codebases, which has following general requirements:
- Benchmarking tests must be provided. This is not the case in the present submission, but the authors have everything they need to add such tests. I suggest the following avenues:
- Demonstrating the exact reproduction of public BDTs/networks on the corresponding datasets. This appears to have been partially done in the tests.
-
Comparing inference speed with the original. A footnote in Ref. [5] claims that PetrifyBDT trees are faster and more resource-efficient than the original. A more rigorous benchmark would be an interesting addition to the manual.
-
At least one example application must be presented in detail. The intended usage is clear and the software appears to meet the requirements. A case study showcasing the package's capabilities would nevertheless be extremely useful as an example for the community to follow. This could also be used as the main benchmarking use case.
-
High-level programming standards must be followed throughout the source code. The code is distributed in the form of a Python package based on setuptools. Tests of the high-level functionality are available, covering ~3/4 of the source lines. Overall, I found the code well written and easy to follow, though parts involving ROOT are naturally more difficult. A higher comment density would sometimes be beneficial.
-
The userguide must properly contextualize the software, describe the logic of its workings and highlight its added value as compared to existing software. The guide provides a historical perspective of the increasing reliance on machine learning in high-energy physics, and of the reinterpretation challenges it entails. The need for reinterpretation could be further contextualized by referring to the FAIR principles [Sci.Data 3 (2016) 1, 160018] and their implementation in machine learning contexts [2212.05081]. The ONNX format has emerged as a the de-facto standard for the exchange of machine learning models. The authors propose an alternative approach for lightweight models, namely direct translation to source code without external dependencies. Such code can have excellent durability characteristics (Fortran 70 code can still be compiled today) at the cost of decreased flexibility, hindering future performance improvements or programmatic inspection of the model. Translation of machine learning models to source code is routinely performed for inference on embedded systems with, e.g., emlearn, onnx2c, or hls4ml. In particular, emlearn's advertised feature set overlaps with the authors' software. This should be addressed in the introduction. PetrifyML originates from PetrifyBDT. As this is version 2.0 of the software, a brief changelog would be a useful addition. The guide provides detailed installation instructions, motivation for each included conversion tool, and a description of known limitations.
-
The software must address a demonstrable need for the scientific community. The software addresses the lack of stable and portable formats to export models trained in several widely-used machine learning frameworks in high-energy physics. This is undeniably needed, though one could regret that ONNX export capabilities are implemented in a third-party package and not upstream (which is of course not always possible). The tools described in Section 4 take ATLAS-specific formats (MVAUtils) as inputs. While this is only relevant for files published by ATLAS, there are MVAUtils files in the wild and using them without relying on ATLAS internal code is a useful addition.
-
The documentation must be complete, including detailed instructions on downloading, installing and running the software. PetrifyML is distributed via the Python package index and can be installed with standard pip commands. The authors provide extensive support for installing only part of the functionality, minimizing the footprint. A complete documentation of all available conversion commands is provided in the documentation. Unfortunately, the documentation does not explain the interface of the emitted code and the user must inspect the result to guess the correct usage. While not particularly difficult, this could be avoided by adding a description in the manual.
I have additional comments for the authors based on my reading of the paper and code. I only consider the first two as essential for publication: - The legal status of the emitted files is unclear. PetrifyML is licensed under version 3 of the GPL and generated files contain code snippets coming from the PetrifyML code. It could be claimed that these files are also covered by the GPL. A clear statement from the authors would be welcome, possibly in the form of a GPL exception similar to GCC's. - I found a crash when converting a GradientBoostingClassifier to C++ (scikit-learn 1.7.2, Python 3.13.7, petrifyml 2.0.0), when the BDT input names are not specified. In this case the attribute gbclassifier.feature_names_in_ does not exist, tripping the code. I implemented a workaround with hasattr() and conversion succeeded. - The package can produce C++ or Python code, which are natural choices for current HEP applications. Once decided, the language can only be changed at the cost of an error-prone manual translation. Guidelines regarding the language(s) to choose for preservation would be useful. In particular, it seems that a plain C interface would be easier to use from other languages. - In some cases, the user is expected to provide the number of inputs and/or classes of the model being converted, which is error-prone. It is unclear whether this requirement is a limitation of the input formats or a shortcoming of petrifyML. - Validation procedures are provided for several conversion tools. The inputs used by these checks are Gaussian-distributed random numbers of various widths, which may not match the expected input range (as noted in Section 4.1.3). For BDTs, a valid range could be determined for each variable based on the cuts used by the trees. - The authors aim to lower the requirements for running petrifyML to a bare minimum. However, the code is not entirely portable across operating systems and some secondary functionality will not work as intended on Windows (due to the use of \ as directory separator) or FreeBSD (which uses clang as the main compiler). The Python version requirement (3.11) excludes RHEL 9 without add-ons. While not essential, I recommend that the authors address these shortcomings. - In C++, the use of templates for the BDT functions leads to duplication of the tree's data for each instantiation. For the SKL BDT in particular, the compiler duplicates the whole switch() statement, dramatically increasing the binary code size. This should preferably be avoided.
Textual comments: 4.1.3: lxplus should be explained 4.1.4: compatability -> compatibility 6: It may seem that the tool is developed specifically for the needs of ATLAS and that other contributions are not welcome.
Requested changes
1- Describe a realistic use case in detail, including a characterization of the performance. 2- Address the overlap with code-generation tools used for embedded inference. 3- Document the interface of the emitted functions. 4- Document the requirements for the lawful use of emitted files in other projects (this does not have to be in the manual, a header in the files would suffice). 5- Fix the conversion bug with GradientBoostingClassifier
Recommendation
Ask for major revision

---

## Round 1 · Referee Report · Anonymous (Referee 2) · 2025-11-1

The referee discloses that the following generative AI tools have been used in the preparation of this report:
ChatGPT-5 was used for correcting spelling and grammar mistakes in the report.
Strengths
1) Well-written article 2) Concise and clear language 3) Examples on how to run each of the scripts 4) Validation available for all scripts
Weaknesses
1) The in-code documentation could be improved 2) There are issues with some of the scripts 3) The exception handling is very limited
Report
The paper presents a new Python package, petrifyML, which enables the conversion of Machine Learning models developed with HEP ML tools into either industry-standard ONNX format or native Python/C++ code. The package is publicly available on GitLab and PyPI, with installation instructions provided in the paper. The installation process is straightforward and allows users to install either the full package or only selected modules.
Currently, the package supports the following conversions:
1) BDTs trained with scikit-learn; .pkl or .job files -> C++ code,
2) BDTs created with the TMVA library; XML files -> C++/Python code,
3) Trees trained with XGBoost/LightGBM within the MVAUtils framework; ROOT files-> ONNX,
4) MLP networks created with TMVA; XML files -> ONNX,
5) Networks saved as lwtn JSON files -> ONNX.
All modes of operation are clearly described in the paper, along with detailed instructions on how to run them and explanations of the available optional arguments. Notably, the authors discuss common issues, e.g. those related to opset and IR version mismatches between ONNX and OnnxRuntime, and offer practical solutions to mitigate them.
A particularly praiseworthy aspect is the inclusion of optional validation for all conversion scripts. In some cases, validation can be triggered automatically by specifying a dedicated flag, while in others, users must manually compile and execute the code. Although this manual step can be slightly inconvenient, it is understandable given the additional dependencies involved. Some converters are not yet fully general; for instance, the TMVA BDT converter does not accept ROOT files, leaving room for future development.
Overall, the package effectively addresses the important issue of preserving and reusing results from experimental analyses, which is of significant relevance to the reinterpretation community. The paper itself is well-structured and clearly written, and I have no requests regarding its text.
However, I have several concerns about the code quality and documentation. The in-code documentation is incomplete, and it should be improved. Moreover, some scripts contain serious issues that may lead to failures or crashes - often due to trivial reasons, such as input files containing a dash ("-") in their names or a trailing comma in a list of parameters - without proper exception handling. Code intended for use by the scientific community must exhibit a high level of usability, robustness, and stability. Therefore, I ask for a major revision.
Requested changes
1) Please improve the in-code documentation. At minimum, every named function (even short ones) in every .py file should include a docstring containing: I) a brief description of the function’s purpose, II) a description of all arguments, and III) a description of the returned value.
2) The link to the project’s README file on the PyPI page is currently broken. Please fix this link so that users can access the documentation directly from PyPI.
3) In the recent version of scikit-learn, the "feature_names_in_" attribute of the GradientBoostingClassifier is created only when X has feature names that are all strings. The converter currently assumes that this attribute is always present, which causes failures. Please modify the code to handle this case properly.
4) In the conversion to C++ code (for both scikit-learn and TMVA models), variable names in the generated code are derived from the output file name. This is unsafe and incorrect. If a user specifies the name of the output file with a white space or a dash, the name of the variable will be incorrect according to C++ standards, and the code will fail to compile. What is even worse, it opens a possibility to inject any code simply by providing it in a filename...
5) When using the --run-validation flag, if compilation of the C++ code fails, no diagnostic message is provided. The program only reports a non-zero exit status, without showing the compiler error output. Please redirect compiler error messages to standard output or provide a clear failure message indicating that the compilation failed and how to diagnose the error.
6) When testing the petrify-tmvabdt scripts with validation, if I provide the wrong --nClasses value, using 3 instead of the real 2, I get the following error "ValueError: operands could not be broadcast together with shapes (3,) (2,) " From this error, it is easy to infer the correct number of classes. This indicates that the issue could be detected programmatically. Please consider implementing exception handling or automatic inference of the correct number of classes.
7) Converter from TMVA MLP to ONNX seems to be the most problematic component. It works for the example file provided in the project repository, but my attempts to use the converter for other files failed. For example, the repository of the CLASS project (https://gitlab-preprod.in2p3.fr/sens/CLASS) contains multiple files in that format, for example, this one: https://gitlab-preprod.in2p3.fr/sens/CLASS/-/blob/4e7493a0c20ce94a78bd9f4860d7bef7a0e01867/DATA_BASES/PWR/MOX_Am/EQModel/MLP_Kinf/TMVARegression_MLP_k_inf.weights.xml It seems that the converter has problems with parsing the "HiddenLayers" option. Files in that repository use a placeholder/variable "N" to define the number of hidden layers, which the converter is not able to parse. Even when the numbers are explicitly provided, the given example will not work. I noticed that the converter cannot parse trailing commas, e.g. in the official example, changing HiddenLayers from (5,10,5) to (5,10,5,) makes it fail. This suggests that the parser is not general enough. If the failure is due to a version mismatch, please specify which TMVA MLP versions are supported. If the problem lies in the parser, please fix it. If the provided example file is non-standard or corrupted, please explain why it is incompatible with your converter.
8) During validation of the petrify-lightweightnn-to-onnx converter, users are required to manually provide four paths to the ONNX and lwtn libraries and header files. It would be helpful if the paper and/or the program’s standard output provided guidance for less experienced users—for example, by showing typical paths (such as those in a default conda installation).
9) Overall, the converters lack sufficient exception handling. The scripts make strong assumptions about input file structures and crash when those assumptions are not met. Improved error handling-especially in the validation components-would make the package significantly more robust and user-friendly.
Recommendation
Ask for major revision

---

## Editorial Decision

unknown